# COVID-19: S-Peptide RBD _484–508_ Induces IFN-γ T-Cell Response in Naïve-to-Infection and Unvaccinated Subjects with Close Contact with SARS-CoV-2-Positive Patients

**DOI:** 10.3390/v15071417

**Published:** 2023-06-22

**Authors:** Michela Murdocca, Gennaro Citro, Eleonora Centanini, Rosalinda Giannini, Andrea Latini, Federica Centofanti, Eva Piano Mortari, Dario Cocciadiferro, Antonio Novelli, Sergio Bernardini, Giuseppe Novelli, Federica Sangiuolo

**Affiliations:** 1Department of Biomedicine and Prevention, University of Rome Tor Vergata, 00133 Rome, Italy; michela.murdocca@uniroma2.it (M.M.); gennaro.citro46@gmail.com (G.C.); eleonora.centanini@gmail.com (E.C.); rosalinda.giannini21@gmail.com (R.G.); latini.andrea@hotmail.com (A.L.); federica.centofanti@gmail.com (F.C.); sangiuolo@med.uniroma2.it (F.S.); 2B Cell Unit, Immunology Research Area, Bambino Gesù Children’s Hospital, IRCCS, 00165 Rome, Italy; eva.pianomortari@opbg.net; 3Medical Genetics, Bambino Gesù Children’s Hospital, IRCCS, 00165 Rome, Italy; dario.cocciadiferro@opbg.net (D.C.); antonio.novelli@opbg.net (A.N.); 4Department of Experimental Medicine, University of Tor Vergata, 00133 Rome, Italy; bernardini@med.uniroma2.it; 5IRCCS Neuromed, 86077 Pozzilli, Italy; 6Department of Pharmacology, School of Medicine, University of Nevada, Reno, NV 89557, USA

**Keywords:** SARS-CoV-2, peptide, IFN-γ, pseudovirus

## Abstract

Despite the availability on the market of different anti-SARS-CoV-2 vaccines, there are still unanswered questions on whether they can stimulate long-lasting protection. A deep understanding of adaptive immune response to SARS-CoV-2 is important for optimizing both vaccine development and pandemic control measures. Among cytokines secreted by lymphocytes in response to viral infection, IFN-γ plays a pivotal role both in innate and adaptive immunity. In this study, we report on 28 naïve-to-SARS-Cov-2-infection and unvaccinated subjects, having reported a close and prolonged contact with COVID-19-positive patients. Samples were tested for defective genetic variants in interferon pathway genes by whole exome sequencing and anti-IFN autoantibodies production was investigated. Subject T-cells were cultured and infected with pseudotype particles bearing the S proteins and in parallel stimulated with two S-peptides designed on the RBD region of the spike protein. Our results showed that one of these peptides, RBD 484–508, induces a significant increase in IFN-γ gene expression and protein production in T-cells, comparable to those obtained in cells infected by SARS-CoV-2 pseudovirus. This work deepens our understanding of immune response and highlights the selected peptide as a reasonable approach to induce broad, potent, and variant concern-independent T-cell responses.

## 1. Introduction

COVID-19 is an acute viral illness caused by the severe acute respiratory syndrome coronavirus 2 (SARS-CoV-2) infection. It has had a dramatic effect on the world’s population, resulting in more than 6.5 million deaths worldwide. As of March 2023, the World Health Organization has confirmed more than 760 million human cases worldwide and 6.8 million (1.1%) deaths (https://www.who.int/emergencies/diseases/novel-coronavirus-2019 (accessed on 20 September 2022).

Immunity is a multifaceted phenomenon. T cell immunity plays a pivotal role in infection control, as well as in long-lasting protection conferred by immune memory, even in the absence of humoral immune responses [1]. Although the pandemic is over, the virus continues to circulate. There is a critical need for foundational knowledge about T cell responses to SARS-CoV-2 because the nature of T cell reaction toward this virus and its implications in COVID-19 pathogenesis are still unclear. Previous studies underlined that cross-reactive CD4+ T cells that recognize severe acute respiratory syndrome coronavirus 2 (SARS-CoV-2) are more commonly detected in the peripheral blood of unexposed individuals compared with SARS-CoV-2–reactive CD8+ T cells. However, the pre-existence of reactive CD8+T cells in the unexposed unvaccinated population and its implications are still controversial and need detailed studies [2].

A recent study, using a series of peptides derived from the viral proteome epitopes, established that unexposed–unvaccinated individuals carry a significant fraction of circulating CD8+T cells reactive to various SARS-CoV-2 proteins. These reactive cells are distributed between memory and naïve T cells [3]. Interferon gamma (IFN-γ) is an antimicrobial cytokine secreted by activated B and T lymphocytes that stimulates several biological responses, mainly involved in host defense and immune surveillance [3]. An increase in IFN-γ production was observed in patients with COVID-19, also prior to the release of antibodies. Therefore, this potent cytokine represents the first important mediator of the innate and adaptive immune responses during the SARS-CoV-2 infection [4].

Even though the typical mechanisms of pre-existing reactivity are not fully understood, it has been linked to cross-reactivity against circulating “common cold” coronaviruses, in regions where it is endemic and in individuals who have not been exposed to SARS-CoV-2. Furthermore, a high level of background activation without any peptide stimulation, in some individuals, may prevent the detection of low-frequency virus-specific T cell responses [5].

The peptide platform represents a preventive approach, which allows the in vivo production of epitope-based vaccines designed with synthetic peptides focusing a specific response versus selected epitopes. By using this approach, specific neutralizing antibodies against selected sequences are produced by the host immune system. Moreover, these short sequences cannot represent immunodominant sites with respect to those stimulated by using the whole protein as antigen. Peptide-based strategy represents an interesting therapeutic approach ensuring efficiency, specificity, and tolerability. Peptide chemical composition makes them highly specific and versatile: they are short sequences that can be easily modified for increasing their half-life [6].

In a previous paper, we demonstrated that spike peptides (S-peptide) are able to induce potent anti-SARS-CoV-2 neutralizing antibody responses and confer significant protection against the SARS-CoV-2 challenge [6].

In this research, our aim was to evaluate in subjects naïve to SARS-CoV-2 infection whether selected S-peptides were able to stimulate the SARS-CoV-2 T cell immunity by inducing functional IFN-γ synthesis, summarizing what occurs in the natural infection. Furthermore, the availability of epitopes suitable for monitoring the T-cell response can be of help in selecting those individuals who need vaccination boosters.

## 2. Materials and Methods

### 2.1. Study Design, Patients, and Informed Consent

From January to March 2021, we enrolled on a voluntary basis twenty-eight unvaccinated individuals (8 men and 20 women, mean age of 43 ± 13.74 years), naïve to SARS-CoV-2 infection, although in close contact with SARS-CoV-2-positive patients. Inclusion criteria comprehended being unvaccinated and reporting a prolonged contact without the use of personal protective equipment with at least one infected and symptomatic subject. Moreover, the SARS-CoV-2 antibody negativity of study subjects (rapid antibody test and/or immunoassay) from 15 to 90 days after exposure was confirmed. Prolonged contact was defined as more than one hour per day for at least 5 days after the beginning of symptoms or as being bed partners for at least three days. None of the volunteers were healthcare professionals, nor suffered from immunodeficiency disorders or from autoimmune disease. Lack of current SARS-CoV-2 infection was confirmed by PCR testing at the time of enrollment. No one reported suffering from chronic diseases (cardiologic, rheumatologic, kidney, pulmonary, autoimmune), diabetes, hypertension, or immunodeficiencies (primary, HIV, organ transplantation/immunosuppressive treatment). All participants provided written informed consent.

### 2.2. Isolation of Blood Cell, Lymphocytes T Culture, and ELISA Assay

The sequences of the two peptides belonging to spike were designed by us, choosing the area comprising the cryptic domain on the RBD sequence and tested in our previous paper (6). Specifically, we have selected and synthesized RBD peptide sequences located within the RBD conserved region (RBD484–508 EGFNCYFPLQSYGFQPTNGVGYQPY; RBD453–476 YRLFRKSNLKPFERDISTEIYQAGS [6]). Firstly, immunoinformatic methodologies identified RBD484–508 and RBD453–476 as highly promising antiviral peptides (T-cell epitopes) to impede the pathogenic process of SARS-CoV-2. Successively, we administered the peptides in vivo in mice, demonstrating their ability to induce potent anti-SARS-CoV-2 neutralizing antibody response. Mice sera conferred significant protection against the SARS-CoV-2 challenge, inhibiting in vitro virus host cell entry [6].

Peripheral blood mononuclear cells (PBMC) from 28 subjects were isolated by density–gradient sedimentation using Ficoll-Paque (GE Healthcare, Chicago, IL, USA) and then were cryopreserved until use. PBMCs were cultivated in RPMI 1640 medium in 96-well plates, supplemented with 10% FBS, 0.2 mM L-glutamine, and 100 units/mL penicillin-streptomycin (Euroclone, Milan, Italy), and activated with PHA (1 μg/mL, Sigma, St. Louis, MO, USA) and IL-2 (100 IU/mL, Sigma, St. Louis, MO, USA) to generate T-cell lines. After five days of culture, cells were divided into four groups in triplicate for each patient. Nothing was added to the cell controls (Ctrl), cells infected with SARS-CoV-2 VSV pseudovirus (S+) and stimulated with spike peptides RBD_484–508_ and RBD_453–476,_ separately, for 24 h. Then, 24 h post-infection/stimulation, cell culture with SARS-CoV-2 VSV-based pseudoparticles and peptides were removed. The culture medium was replenished and analyzed following 24 h and 4 days by using an enzyme-linked immunosorbent assay (ELISA) to detect and quantify -IFN gamma antibody using a kit purchased from Cohesion Biosciences (catalogue CK1190).

For the detection of IgG anti-IFN, an ELISA test was performed on plasma, as previously published [7]. Optical density (OD) measurement was assessed.

### 2.3. Gene Expression Analyses

Total RNAs from lymphocytes were extracted by Trizol Reagent (Invitrogen Life Technologies Corporation, Carlsbad, CA, USA) following the manufacturer’s instructions. Treatment with DNase I-RNase-free (Ambion, Life Technologies Corporation, Foster City, CA, USA) was used to eliminate genomic DNA contamination from total RNA samples. One μg of RNA was reverse transcribed with the High-Capacity cDNA Archive kit (Life Technologies Corporation, Foster City, CA, USA) and used in RT-PCR and RT-qPCR. mRNAs were measured by SYBR Green (Life Technologies Corporation, Foster City, CA, USA) using the following primers: 5′-TCATCAGCAATGCCTCCTG and 3′-GGCCATCACGCCACAGTT for GADPH as reference gene; 5′-GCATCCAAAAGAGTGTGGAG and 3′-GACAGTTCAGCCATCACTTGG for IFN-gamma. The 2-(ΔCt) and comparative ΔΔCt methods were used to quantify relative gene expression levels.

### 2.4. Whole Exome Sequencing

Genomic DNA was extracted from peripheral blood samples using standard procedures and Qiagen blood DNA mini-Kit (Qiagen, Hilden, Germany). Library preparation and whole exome capture were performed by using the Twist Human Core Exome Kit (Twist Bioscience, South San Francisco, CA, USA), according to the manufacture’s protocol, and sequenced on the Illumina NovaSeq 6000 platform. The BaseSpace pipeline (Illumina, Inc., San Diego, CA, USA) and the TGex software (LifeMap Sciences, Inc., Alameda, CA, USA) were used for variant calling and annotation, respectively. Sequencing data were aligned to the hg19 human reference genome. Based on the guidelines of the American College of Medical Genetics and Genomics, a minimum depth coverage of 30× was considered suitable for analysis. Variants were examined for coverage and Qscore (minimum threshold of 30) and visualized by the Integrative Genome Viewer (IGV).

### 2.5. Statistical Analyses

All the experiments were performed in technical duplicates, and data were analyzed using GraphPad Prism 8 and the SPSS program, version 25 (IBM Corp, Armonk, NY, USA). The difference between groups was tested by a paired Student *t*-test and one-way ANOVA test with post-hoc Tukey HSD (Honestly Significant Difference). Values displayed in the figures represent the means of three independent experiments ± standard deviation (SD). Statistical significance was established at *p* < 0.05.

## 3. Results

### Peptide RBD_484–508_-Induced IFN-γ T Cell Response

T cells were cultured for up to 5 days, according to the experimental procedures described in the material and methods sections. The supernatants were collected at 24 h and 4 days post-infection/stimulation and quantified by ELISA protocols to detect IFN gamma antibody, according to the manufacturer’s recommendations. The IFN-γ secreted in the medium spanned across a high range of values and, most importantly, the high heterogeneity in the IFN-γ production was found not only in the infected (S+) and stimulated (RBD _484–508_ and RBD _453–476_) cells, but also in the controls (Ctrls) (data not shown). Considering the variability among groups, the samples of stimulated cells were sorted based on increasing values of ODs measured by ELISA in the Ctrls and roughly divided into subgroups from one to three (Table 1).

Group 1 includes a low basal level of IFN-γ and it was considered for downstream analysis. This is because it is not possible to appreciate and evaluate the stimulating activity of either the virus or the peptides in samples in which the control had high levels of IFN-γ.

For all the subjects, the absence of defective genetic variants in genes coding for protein of the interferon pathway, including IFNAR1, IFNAR2, IRF3, IRF7, IRF9, STAT1, STAT2, TBK1, TICAM1, TLR3, TRAF3, and UNC93B1, was performed by whole exome sequencing (Table 2), as well as the absence of the production of anti-IFN autoantibodies (data not shown) [7]. The eight variants reported, even if rare, were classified as benign or likely benign. Previous studies have indeed assessed that genetic mutations and autoantibody production could underlie a reduced interferon response, lowering the ability of the body to respond to viral infections [7].

In accordance with the above concepts, Group 1 (9 individuals of 28), characterized by samples with a low basal level of IFN-γ, was considered more suitable for the stimulating activity of both the virus and peptides.

In fact, the concentrations of IFN-γ in Group 1 were more homogeneous if compared with Groups 2 and 3, expressing a higher rate of IFN-γ.

The quantitative analyses of IFN-γ, conducted by ELISA on the culture medium of T-lymphocytes, showed an increasing trend of the cytokine in the group stimulated for 24 h with the peptide RBD_484–508_ with respect to the control group, but the values were not statistically significant (data not shown). Instead, after 4 days of cell culture, the trend became statistically significant and the values were comparable with those obtained in the S+ group, as expected (Figure 1A). On the other hand, the other peptide RBD_453–476_ did not show any elicitation of the cellular response, probably due to the absence of linear epitopes recognized by the MHC (major histocompatibility complex). The processing of the antigen or presentation of the antigen (APM, antigen processing machinery) is that biological mechanism implemented by each cell of the organism capable of presenting an antigen (on MHC I or MHC II), which allows the expression of antigenic peptides on molecules of the major histocompatibility complex. Furthermore, processing is of great importance to select those antigenic determinants (8–11 aa) that bind lymphocyte receptors with greater affinity. Antigenic proteins can possess innumerable determinants (or epitopes), but only some of these have the characteristics that allow the binding with MHC molecules. These epitopes are also called immunodominant epitopes and their exposure as single peptides serves to instruct T lymphocytes which part of the antigen to bind to increase their responsiveness.

To confirm the effect of peptide RBD_484–508_ on immune response and to support its capacity to emulate the virus stimulation, we compared the levels of IFN-γ mRNA in T cells after 4 days from pseudovirus infection and peptide stimulation, with respect to the control.

We observed, by RT-qPCR, that both pseudo-SARS-CoV-2 (S+) and peptide (RBD_484–508_) induced a significant increase in IFN-γ expression, indicating a similar response (Figure 1B). These molecular data are consistent with the data of IFN-γ concentration in the supernatant measured after stimulation with the pseudovirus or the peptide designed in the SPIKE’s conserved region of RBD. Thus, this peptide appears important for the induction of IFN-gamma in T cells.

## 4. Discussion

COVID-19, the highly contagious viral disease caused by the severe acute respiratory syndrome coronavirus 2 (SARS-CoV-2), has spread worldwide with millions of cases, determining mild to severe respiratory infections and more than 6 million deaths to date (https://www.who.int/emergencies/diseases/novel-coronavirus-2019 (accessed on September 2022). To face the rapid emergence of the pandemic, a better understanding of T-cell response mechanisms to the virus is crucial for developing optimal immunotherapies and vaccines.

Nevertheless, the data regarding the nature of T-cells and the role that they play in COVID-19 protection and pathogenesis is incomplete and sometimes conflicting. To date, there have been few studies analyzing SARS-CoV-2-specific T cell responses and their role in disease progression [8]. Investigating the adaptive immune response against SARS-CoV-2 could contribute to a better understanding of native immune response.

IFN-γ is a pleiotropic cytokine, a key player in driving cell-mediated immunity, capable of orchestrating numerous protective functions to heighten immune responses in infections and other pathologies.

Our study focused on T-cell responses in twenty-eight subjects naïve to SARS-CoV-2 infection (8 men and 20 women, mean age of 43 ± 13.74 years). All T-cell lines were infected with SARS-CoV-2 VSV-based pseudoparticles and stimulated with spike peptides for 24 h. Due to the high variability among groups, the cohort was stratified in three groups and only Group 1, including individuals expressing low basal level of IFN-γ, was considered for downstream analysis.

In most current vaccines, the extent of protection and its duration remain uncertain, especially regarding the emergence of new virus variants of concern. Antibody response, previously triggered by current vaccines, could not protect against severe progression, implicating the importance of T cell responses. Thus, this epitope strategy (small antigen to overcome immunodominant problems in the same protein) depicts an excellent approach to be explored for a vaccine, considering also that selected epitopes are common to all SARS-CoV-2 strains, as deduced by phylogenetic analysis [9]. The conserved epitopes allow generating immunity that is not only cross-protective over coronaviruses, but also relatively resistant to ongoing virus evolution, as well as future pandemics.

In our previous study [6], we identified RBD_484–508_ and RBD_453–476_ peptide to impede the pathogenic process of SARS-CoV-2, using immune informatics methodologies. Our data in vitro and in vivo demonstrated our capacity to induce potent anti-SARS-CoV-2 neutralizing antibody responses, eliciting a significant protection against SARS-CoV-2 challenge.

The immune response varies consistently across individuals, and it can be influenced by several factors, including genetics, environment, and clinical history. We have verified the absence of defective genetic variants in the interferon pathway, as well as the absence of the production of anti-IFN autoantibodies.

We observed a variable basal production of IFN-γ in all samples, probably due to cross-reactive CD4+ and CD8+ memory T cells due to previous exposure to seasonal coronaviruses, which could have a significant impact on the immune reactivity to future SARS-CoV-2 infection.

To appreciate the stimulating activity of the peptides, we decided to consider the group of samples with a low basal level of IFN-γ. The quantitative analyses conducted by ELISA on the culture medium of T-lymphocytes exhibited an increase in the central antiviral mediator in lymphocytes, in cells stimulated with peptides RBD_484–508_ after 4 days of cell culture, when compared with the controls. This trend is similar to those shown in cells infected by pseudovirus (S+), as expected, and resulted as statistically significant.

Peptide-based vaccines, which represent one of the most potent vaccine platforms, offer exclusive advantages over classical vaccines that use whole organisms or proteins and have great potential for personalized immunotherapy. The platform also allows the production of specific antibodies against the S protein of SARS-CoV-2. In particular, using peptides focused on the conserved RBD region may allow structure-based design not only of a SARS-CoV-2 vaccine but also of cross-protective antibody response against future coronavirus epidemics and pandemics. This peptide may well serve as a complementary vaccine to stimulate T cell immunity, particularly in elderly and immunocompromised individuals with defective ability for sufficient immune responses after SARS-CoV-2 vaccination with currently approved vaccines.

In conclusion, the results of this study indicated that RBD_484–508_ peptide is a candidate for the induction of a significant SARS-CoV-2 T cell immunity, showing a favorable safety profile and inducing broad, potent, and variant concern response.

We are aware that this study presents three limitations: the low number of individuals, due to the difficulty in recruiting unvaccinated people, the selection of epitopes belonging to only one of the viral antigens tested, and the low number of individuals included in the downstream analysis.

## Figures and Tables

**Figure 1 viruses-15-01417-f001:**
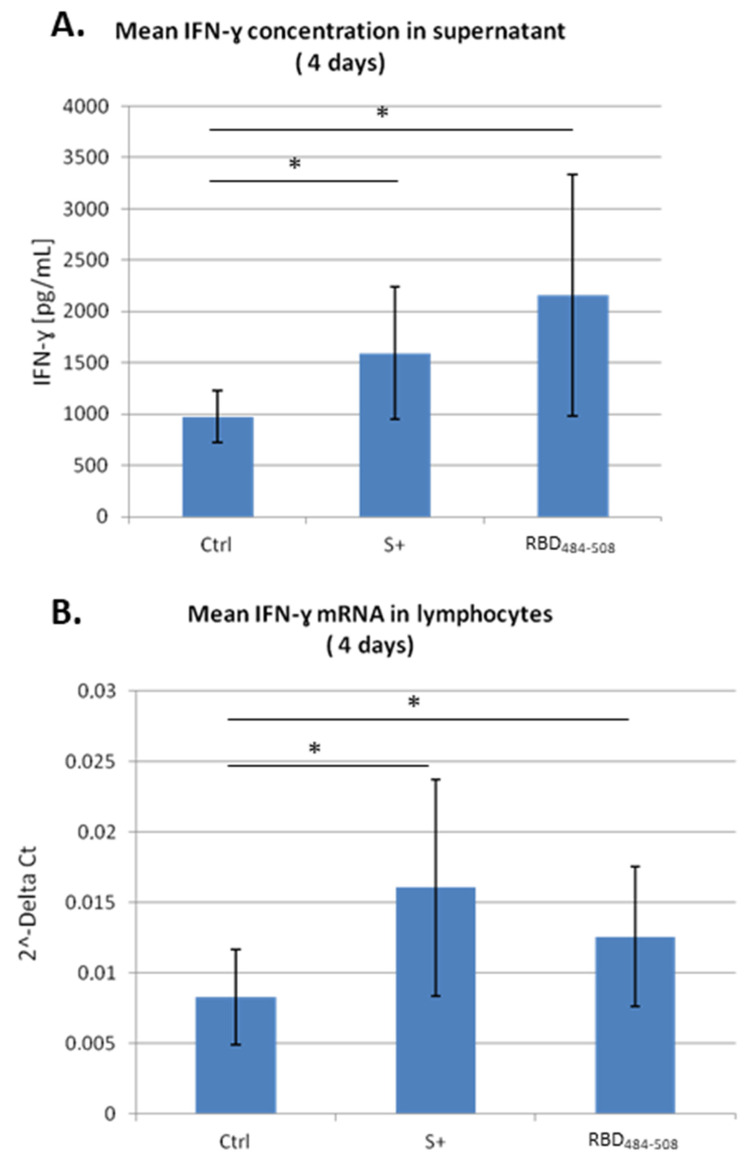
(**A**): IFN-γ concentration [pg/mL] in the supernatant and (**B**) mRNA lymphocyte expression level 4 days after pseudovirus infection (S+) or following RBD_484–508_ peptide treatment. Values displayed in the figures represent the means of three independent experiments ± standard deviation (SD). * *p* < 0.05 by Anova test.

**Table 1 viruses-15-01417-t001:** List of the 28 samples with the corresponding values of optical density (OD) of the controls (Ctrls) sorted in ascending order and subdivided into three groups.

Subject ID	OD (CTRLs)		
# 87	0	Group 1	
# 43	0.04		
# 49	0.106		
# 45	0.14		
# 78	0.175		OD < 0.4
# 86	0.217		
# 64	0.258		
# 80	0.273		
# 65	0.294		
# 88	0.46	Group 2	
# 18	0.63		
# 71	0.7		
# 68	0.707		
# 63	1.042		0.4 < OD < 1.7
# 67	1.08		
# 72	1.5		
# 42	1.533		
# 69	1.6		
# 79	1.659		
# 77	1.76	Group 3	
# 73	1.98		
# 66	2.066		
# 47	2.179		
# 76	2.57		OD > 1.7
# 89	2.94		
# 70	3.1		
# 82	3.11		
# 81	4		

**Table 2 viruses-15-01417-t002:** Variant classification identified in 8 of 28 individuals.

Chr.	Pos.	Gene	Variant Type	dbSNP	gnomAD	ClinV.	ACMG	Variant
chr19	50165329	IRF3	synonymous	.	0	.	LB (BP4, BP7, PM2)	NM_001571.6 c.G858T:p.L286L
chr19	4817446	TICAM1	missense	.	0	.	LB (BP4, BP1, PM2)	NM_182919.3 c.C944T:p.P315L
chr12	56750350	STAT2	synonymous	rs540438595	9.55 × 10^−5^	.	LB (BP4, BP6, BP7, PM2)	NM_005419.4 c.G6A:p.A2A
chr21	34721474	IFNAR1	missense	rs141681155	0.0005	.	LB (BP4, BP1, PM2)	NM_000629.3 c.C866T:p.T289I
chr19	50168062	IRF3	synonymous	.	3.99 × 10^−6^	.	LB (BP4, BP7, PM2)	NM_001571.6 c.C34T:p.L12L
chr21	34625037	IFNAR2	missense	rs147496374	0.0049	B	B (BS1, BS2, BP4, BP6, BP1	NM_001289125.1 c.C611G:p.T204R
chr19	4816598	TICAM1	missense	rs150224968	0.0006	LB	B (BP4, BP6, BP1, PM2)	NM_182919.3 c.G1792T:p.G598W
chr19	50162953	IRF3	missense	rs141490768	0.0008	.	B (BS1, BS2, BP4, BP1)	NM_001197122.2 c.G1252A:p.A418T

Chromosome (Chr), Genome Aggregation Database (gnomAD), allelic frequency (AF), putative loss of function (pLOF), position (Pos), Single Nucleotide Polymorphism Database (dbSNP), Clinvar (ClinV), American College of Medical Genetics (ACMG). A threshold of 0.01 as allele frequency in GnomAD was set.

## Data Availability

Data are available on demand.

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
