# Peer review of "COVID-19: S-Peptide RBD 484–508 Induces IFN-γ T-Cell Response in Naïve-to-Infection and Unvaccinated Subjects with Close Contact with SARS-CoV-2-Positive Patients"

_viruses, 2023, doi:10.3390/v15071417_

Round 1

Reviewer 1 Report

The design of the study is good, and the paper is concise and relevant. The variables are well defined and measured appropriately. The study methods are valid and reliable. There are enough details provided in order to replicate the study.

The data is presented in an appropriate way. The text in the results add to the data and it is not repetitive. Statistically significant results are clear. It is clear which results are with practical meaning. Results are discussed from different angles and placed into context without being overinterpreted. However, there are some issues:

Major:

1. The hypothesis is not well-covered in the introduction.

2. The conclusion should be more strong.

Minor:

1. The authors should declare the end of the pandemic.

2. How did the authors confirm the SARS-CoV-2 naive status of the patients? Have you performed antibodies and cellular immunity against the virus?

3. Line 263 correct the word "ab-sence"

The English style and language are good.

Author Response

Reviewer 1

The design of the study is good, and the paper is concise and relevant. The variables are well defined and measured appropriately. The study methods are valid and reliable. There are enough details provided in order to replicate the study.

The data is presented in an appropriate way. The text in the results add to the data and it is not repetitive. Statistically significant results are clear. It is clear which results are with practical meaning. Results are discussed from different angles and placed into context without being overinterpreted. However, there are some issues:

Major:

The hypothesis is not well-covered in the introduction.

We further stressed the hypothesis within the introduction

The conclusion should be more strong.

We have reworked the conclusion trying to make it more incisive and clear to the reader

Minor:

The authors should declare the end of the pandemic.

We did it

How did the authors confirm the SARS-CoV-2 naive status of the patients? Have you performed antibodies and cellular immunity against the virus?

the negativity to SARS-CoV-2 was assessed by rapid antibody test and/or immunoassay from 15 to 90 days after exposure to infected people.

Line 263 correct the word "ab-sence"

We did it

Reviewer 2 Report

Manuscript viruses-2445511

Manuscript shows results supporting the contention that a peptide (RBD 484–508) that resides in the spike of SARS-Cov-2 may induce a significant increase in the production of gamma interferon by T-cells. Despite its limited scope, the manuscript appears publishable, provided it is substantially edited. Some proposed changes are of a scientific nature, and should be addressed before publication (e.g., the statical treatment of the data; the lack of support for the conclusions).  Other type is more of a “cosmetic” nature, which I believe may improve clarity.  Both types are listed in detail below.

Line 25: What does “designed” mean in this context?

Lines 43-5: The sentence appears Incomplete: “Previous studies on healthy unexposed individuals have indicated mainly CD4+T.” Relative to what other cells?  Studies have shown that cells reactive to SARS-CoV-2 are mainly CD4+T lymphocytes? Or, studies have shown the predominance of “CD4+T cells reactive to SARS-CoV-2”? Somewhat explained by following lines (45-7, but not quite). Clarify.

Line 67: The word “acquired” perhaps is not the best descriptor of the research reported in the manuscript. Interferon production by T cells was induced by “infection” by a pseudovirus. Or in what other way the response to the peptides resembled “that one acquired by natural infection.”? Consider the use of an alternative wording.

Lines 78-9: The SARS-CoV-2 antibody negativity of study subjects (more straightforward language than “negativity to specific antibody testing”) was certainly assessed, BUT (a more important fact) confirmed. Consider revising the wording.

Lines 105-7: Individuals were not “infected with SARS-CoV-2 VSV-based pseudoparticles (pseudovirus?) (S+),” nor “stimulated with spike peptides RBD 484–508 and RBD 453–476.” Their cells were. Wording may create confusion. I would suggest a change. Were cells exposed to a mixture of the two peptides (RBD 484–508 and RBD 453–476). Clarify. This is critical: three groups? It would help to reword indicating what was ADDED to each culture/group of cultures; was something added to the controls?

Line 109: “were removed,” instead of have been removed.” What was removed, the supernatant (line 146)? What does “replaced” mean? Culture medium was replenished?

Lines 111-2: To detect, quantify interferon (IFN, INF-typo?) gamma or anti-interferon gamma (antibodies)? Apparently, both. This should be properly explained. These are critical experimental details. Cohesion (company name)? Where was “IgG anti-INF (IFN)” detected?

Line 145: What 5 days? Material and Methods indicate 7 days in line 105. Explain.

Lines 147-8: See comment to lines 111-2 (“anti-Human gamma IFN?).

Line 152: What data are not shown? At least some data are shown in Table 1, correct? Clarify.

Lines 156-7: What treatment the control cultures received, which show the OD values presented in Table 1? As indicated above, description of treatment (stimulation) of cultures with pseudovirus and peptides in Material and Methods (lines 105-7) does not include treatment of Controls. What does an OD value represent? This should be explained here or, even better, in Materials and Methods.

Line 172: Is it the intention of the authors to insert Table 2 at this level of the manuscript? There is an apparent contradiction between the Title of the table and lines 164-7, in that it is unclear whether the analysis of the absence of defective genetic variants in genes encoding for proteins of the interferon pathway was performed (and presented) for all the 28 individuals or only for the 9 individuals of Group 1.

Line 177-8: I am not clear what the purpose of these lines is, which appear redundant: if the concentrations are more consistent (when?), they are also more homogeneous. Higher rate relative to what? Explain.

Line 181: Where in Materials and Method the study of the induction of interferon by a single peptide is described (see comment to lines 105-7)?

Line 185: What is the meaning of “any elicitation of the cellular response to an increased IFN- gamma production”? Do you mean peptide RBD453–476 did not induce the production of any gamma interferon in the cell cultures? Reword accordingly.

Line 202: What do you mean by “the first protection”?

Lines 204-6: “Our results”? What results? Do you mean that all the results point to the fact that peptide RBD484–508, which lies in the conserved region of SARS-Cov-2 Spike appears important for the induction of gamma interferon? Why don’t you say it in a crisper, more direct, more comprehensible way?

Table 2 basically doesn’t make any sense, even with the inclusion of what appears to be footnotes explaining the columns’ content (lines 211-3). Does each line represent an individual? If not, what? Who are the carriers of the variant? Why does the table have two different shapes (top and bottom), sets of information (with some common data, such as “Chromosome” and “Gene”)? If the authors want to keep the table, it should be thoroughly re-done and properly explained.

What is the relation between OD (Table 1) and pg/mL (Figure 1a)? What are the units in the y-axis in 1b? Explain. This is important experimental information.

Lines 218-21: See comment to lines 105-7. Also, establishing statistical significance of the differences using multiple t-tests is incorrect (multiplicity problem). Although authors indicate that ANOVA was used (section 2.5), there is no indication of this in Figure 1. ANOVA plus post-hoc comparisons is the right alternative to the two t-tests used per panel, although perhaps the best approach would be the use of the Dunnett test (treatments vs. a control). This should be corrected.

Line 238: The study “focused” (past tense). Results of a finished study are presented.

Line 251: What are these “immunodominant problems”?

Line 254: What is the relevance, in the context of COVID-19 protection, of this cross-protection over coronaviruses (vs. SARS CoV-2 strains-variants)?

Line 257: Expression “antiviral peptides” is unusual, in the context of what I believe is an epitope. Consider rewording.

Lines 265-8: See comment to line254. Apparently cross-protective immunity from other coronavirus was not very efficient to protect from primary infection with SARS CoV-2. Why should it have a “significant impact” against subsequent infections? What data (in this manuscript) do indicate this “variable basal production of gamma interferon in all samples,” probably due to exposure to non- SARS CoV-2 coronavirus? Explain.

Line 271: “Highest”, relative to what, if only one group (Group 1) was studied? Are the authors talking about Figure 1a? Higher that S+, but not statistically significant (see comment to lines 218-21)?

Line 273: Why, “however,” if the results were those expected?

Line 275: Drop “in sum.” Another limitation (now three, not two) of the study is the low number of individuals included in the “downstream analysis” (nine, in Group 1; line 243). Say it.

Lines 277- 9: Conclusion is very broad and, as far as I can tell, unsupported by the findings in the manuscript. How did the authors assess the “safety profile” of peptide RBD484–508? How did they establish the magnitude of the extension and of the intensity of the response to the different variants of SARS CoV-2 based on exposure to the peptide? In relation to what?

Line 22: by (a means of something) through (in relation to a process)? Choose one.

Line 42: Comma before “because”

Line 151: “either by pseudovirus or peptides” is redundant.

Lines 230-1: Sentence apparently incomplete: the nature and degree of T-cells, what? The nature of T-cells that participate, and the degree to which they do it? The evidence in support of the role that the T-cells play … and the knowledge of the type involved? Reword.

Author Response

Reviewer 2

Manuscript shows results supporting the contention that a peptide (RBD 484–508) that resides in the spike of SARS-Cov-2 may induce a significant increase in the production of gamma interferon by T-cells. Despite its limited scope, the manuscript appears publishable, provided it is substantially edited. Some proposed changes are of a scientific nature, and should be addressed before publication (e.g., the statical treatment of the data; the lack of support for the conclusions).  Other type is more of a “cosmetic” nature, which I believe may improve clarity.  Both types are listed in detail below.

Line 25: What does “designed” mean in this context?

By using immunoinformatics methodologies, RBD484–508 and RBD453–476 were identified as highly promising antiviral peptides. Specifically, we have selected and synthetized peptide sequences located within the RBD conserved region. Infact four 25-residue sequences (comprising amino-acids 322–508), were predicted as highly potential antiviral ones (T-cell epitopes).

Lines 43-5: The sentence appears Incomplete: “Previous studies on healthy unexposed individuals have indicated mainly CD4+T.” Relative to what other cells?  Studies have shown that cells reactive to SARS-CoV-2 are mainly CD4+T lymphocytes? Or, studies have shown the predominance of “CD4+T cells reactive to SARS-CoV-2”? Somewhat explained by following lines (45-7, but not quite). Clarify.

I agree with the reviewer, the text has been corrected.

Line 67: The word “acquired” perhaps is not the best descriptor of the research reported in the manuscript. Interferon production by T cells was induced by “infection” by a pseudovirus. Or in what other way the response to the peptides resembled “that one acquired by natural infection.”? Consider the use of an alternative wording.

As correctly suggested, we change the verb

Lines 78-9: The SARS-CoV-2 antibody negativity of study subjects (more straightforward language than “negativity to specific antibody testing”) was certainly assessed, BUT (a more important fact) confirmed. Consider revising the wording.

We rephrased the sentence

Lines 105-7: Individuals were not “infected with SARS-CoV-2 VSV-based pseudoparticles (pseudovirus?) (S+),” nor “stimulated with spike peptides RBD 484–508 and RBD 453–476.” Their cells were. Wording may create confusion. I would suggest a change. Were cells exposed to a mixture of the two peptides (RBD 484–508 and RBD 453–476). Clarify. This is critical: three groups? It would help to reword indicating what was ADDED to each culture/group of cultures; was something added to the controls?

We agree with these observations and modify the text

Line 109: “were removed,” instead of “have been removed.” What was removed, the supernatant (line 146)? What does “replaced” mean? Culture medium was replenished?

We modified the sentence

Lines 111-2: To detect, quantify interferon (IFN, INF-typo?) gamma or anti-interferon gamma (antibodies)? Apparently, both. This should be properly explained. These are critical experimental details. Cohesion (company name)? Where was “IgG anti-INF (IFN)” detected?

We better explained in the text, as suggested

Line 145: What 5 days? Material and Methods indicate 7 days in line 105. Explain.

You are right: 5 days

Lines 147-8: See comment to lines 111-2 (“anti-Human gamma IFN?).

We have corrected again

Line 152: What data are not shown? At least some data are shown in Table 1, correct? Clarify.

Data not shown refers to ctrl only We specify within the text

Lines 156-7: What treatment the control cultures received, which show the OD values presented in Table 1? As indicated above, description of treatment (stimulation) of cultures with pseudovirus and peptides in Material and Methods (lines 105-7) does not include treatment of Controls. What does an OD value represent? This should be explained here or, even better, in Materials and Methods.

The control did not receive any treatment. In table 1 are shown only stimulated cells. OD value represents optical density

Line 172: Is it the intention of the authors to insert Table 2 at this level of the manuscript? There is an apparent contradiction between the Title of the table and lines 164-7, in that it is unclear whether the analysis of the absence of defective genetic variants in genes encoding for proteins of the interferon pathway was performed (and presented) for all the 28 individuals or only for the 9 individuals of Group 1.

The legend to table 2 was moved on the bottom of the table 2. WES was done for all 28 individuals. We have corrected in the text

Line 177-8: I am not clear what the purpose of these lines is, which appear redundant: if the concentrations are more consistent (when?), they are also more homogeneous. Higher rate relative to what? Explain.

We agree with the revision and thus we have corrected the phrase.

Line 181: Where in Materials and Method the study of the induction of interferon by a single peptide is described (see comment to lines 105-7)?

We have now specified in materials and methods that cell stimulation was done separately with each peptide

Line 185: What is the meaning of “any elicitation of the cellular response to an increased IFN- gamma production”? Do you mean peptide RBD453–476 did not induce the production of any gamma interferon in the cell cultures? Reword accordingly.

You are right, it is not clear, thus we have corrected the phrase

Line 202: What do you mean by “the first protection”?

We partially delete “first protection” phrase, trying to clarify the concept

Lines 204-6: “Our results”? What results? Do you mean that all the results point to the fact that peptide RBD484–508, which lies in the conserved region of SARS-Cov-2 Spike appears important for the induction of gamma interferon? Why don’t you say it in a crisper, more direct, more comprehensible way?

We have specified in the text: “our molecular results”. We have better clarified the  concept, as suggested.

Table 2 basically doesn’t make any sense, even with the inclusion of what appears to be footnotes explaining the columns’ content (lines 211-3). Does each line represent an individual? If not, what? Who are the carriers of the variant? Why does the table have two different shapes (top and bottom), sets of information (with some common data, such as “Chromosome” and “Gene”)? If the authors want to keep the table, it should be thoroughly re-done and properly explained.

We have modified the table, deleting the upper part, but also specifying  within the text that the 8 variants identified by WES are benign and thus without any pathological effect.

What is the relation between OD (Table 1) and pg/mL (Figure 1a)? What are the units in the y-axis in 1b? Explain. This is important experimental information.

The OD in Table 1 correspond to pg/ml in figure 1a, whose median value is 1000pg/ml and  belongs to basal Value (Ctrl).

In the y-axis in 1b, the mRNA levels are reported as 2^-∆Ct. It is a simple formula used in order to calculate the gene expression of samples when performing a qPCR. ∆Ct is the difference in Ct values for our gene of interest and our housekeeping gene for a given sample. Then, we transformed the values ​​into a logarithmic scale (2^-∆Ct) before undertaking statistical analysis.

Lines 218-21: See comment to lines 105-7. Also, establishing statistical significance of the differences using multiple t-tests is incorrect (multiplicity problem). Although authors indicate that ANOVA was used (section 2.5), there is no indication of this in Figure 1. ANOVA plus post-hoc comparisons is the right alternative to the two t-tests used per panel, although perhaps the best approach would be the use of the Dunnett test (treatments vs. a control). This should be corrected.

The difference between groups was tested by a paired one-way ANOVA test. We have corrected the caption of the figure 1. We also tested the Dunnett test, as suggested by the reviewer, obtaining results comparable to those with the ANOVA test.

Line 238: The study “focused” (past tense). Results of a finished study are presented.

The verb has been corrected

Line 251: What are these “immunodominant problems”?

It is known that there is bias and immunodominance (ID) in the immune response,

which sometimes reduces efficacy. The phenomenon of immunodominance often involves

variations in immunogenicity even between different sites of the same antigen. On the

other hand, it is known that in large antigens, this phenomenon also occurs as a function of

individual polymorphisms [57–60].

Line 254: What is the relevance, in the context of COVID-19 protection, of this cross-protection over coronaviruses (vs. SARS CoV-2 strains-variants)?

This study has been perfomed during the first period of the pandemic when the original strain was expanding across the population and no Spike variants occurred yet. Anyway the epitopes located within the cryptic Spike region are filogenetically conserved and not prone to variation. The amino acid sequences of the oligopeptides reflect the epitopes belonging to the cryptic zone of S protein, which cannot be mutated under penalty of non-functionality of the protein itself (Bao-zhong Zhang, Cell Research 2020; Ming Zheng and Lun Song, Cellular & Molecular Immunology 2020;  Syed Faraz Ahmed, Viruses 2020; Meng Yuan, Science 2020)

Line 257: Expression “antiviral peptides” is unusual, in the context of what I believe is an epitope. Consider rewording.

We have deleted this part of the phrase

Lines 265-8: See comment to line254. Apparently cross-protective immunity from other coronavirus was not very efficient to protect from primary infection with SARS CoV-2. Why should it have a “significant impact” against subsequent infections? What data (in this manuscript) do indicate this “variable basal production of gamma interferon in all samples,” probably due to exposure to non- SARS CoV-2 coronavirus? Explain.

Regarding the association between “variable basal production of gamma interferon” and previous infections by seasonal coronavirus we refer to literature data (Sette, A. & Crotty, S. Pre- existing immunity to SARSCoV- 2: the knowns and unknowns. Nat. Rev. Immunol.2020). Moreover our cohort showing a “variable basal production of gamma interferon” was composed by Naïve to Infection and Unvaccinated Subjects with Close Contacts with SARS-CoV-2 Positive Patients suggesting a protection from SARS-CoV-2 infection

Line 271: “Highest”, relative to what, if only one group (Group 1) was studied? Are the authors talking about Figure 1a? Higher that S+, but not statistically significant (see comment to lines 218-21)?

You are right: both S+ and peptide are equally significant, thus we have rephrased

Line 273: Why, “however,” if the results were those expected?

“However” was deleted

Line 275: Drop “in sum.” Another limitation (now three, not two) of the study is the low number of individuals included in the “downstream analysis” (nine, in Group 1; line 243). Say it.

We have deleted “in sum” and reformulated the phrase introducing the third limitation

Lines 277- 9: Conclusion is very broad and, as far as I can tell, unsupported by the findings in the manuscript. How did the authors assess the “safety profile” of peptide RBD484–508? How did they establish the magnitude of the extension and of the intensity of the response to the different variants of SARS CoV-2 based on exposure to the peptide? In relation to what?

The conclusions has been integrated

The safety of peptide RBD484–508was tested on a cohort of mice to whom the peptide has been administered.

The immunization regimen employed did not induce detectable pathological effects on treated animals (weight loss, loss of appetite, reduced mobility) either during or at the end of vaccinations (Murdocca et al, Viruses 2021) Neutralizing activity of IgG elicited by spike RBD peptides was tested on  VeroE6 cells incubated with pseudotypes particles bearing the S proteins of the indicated Vsvpp.SARS-2-S WT or variants

(i.e., D614G, B.1.1.7, P1, B.1.351), showing a relevant and statistically significant efficiency of infection inhibition (Murdocca et al, Viruses 2021).

Line 22: by (a means of something) through (in relation to a process)? Choose one.

We have corrected

Line 42: Comma before “because”

We have introduced the comma

Line 151: “either by pseudovirus or peptides” is redundant.

We have corrected

Lines 230-1: Sentence apparently incomplete: the nature and degree of T-cells, what? The nature of T-cells that participate, and the degree to which they do it? The evidence in support of the role that the T-cells play … and the knowledge of the type involved? Reword.

You are right. The phrase has been reformulated

Round 2

Reviewer 1 Report

The authors addressed all the issues and the paper has been improved significantly.

Author Response

Thank you very much for your suggestions

Reviewer 2 Report

Almost ready to go, with an important exception:

Line 126: What is being detected using ELISA is interferon gamma (NOT anti-interferon gamma) by ELISA, using a kit purchased from Cohesion Biosciences (Catalogue CK1190 or CK1191) correct? This should be corrected before publication.

Lines 164-5: See comment to line 126

Line 239: I would like to see in writing what post-hoc comparison was used to test for significance of the differences (e.g., Tukey?)

Line 124: were removed; was replenished.

Author Response

Reviewer 2

Line 126: What is being detected using ELISA is interferon gamma (NOT anti-interferon gamma) by ELISA, using a kit purchased from Cohesion Biosciences (Catalogue CK1190 or CK1191) correct? This should be corrected before publication.

Yes, we have modified within the text

Lines 164-5: See comment to line 126

Yes, we have modified within the text

Line 239: I would like to see in writing what post-hoc comparison was used to test for significance of the differences (e.g., Tukey?)

It is Tukey. We have added within the text

Comments on the Quality of English Language

Line 124: were removed; was replenished.

The verbs have been changed